# Nanoscale Phytosomes as an Emerging Modality for Cancer Therapy

**DOI:** 10.3390/cells12151999

**Published:** 2023-08-04

**Authors:** Ahmad Kadriya, Mizied Falah

**Affiliations:** 1Medical Research Institute, The Holy Family Hospital Nazareth, Nazareth 1641100, Israel; ahmadkadriya@gmail.com; 2Azrieli Faculty of Medicine, Bar-Ilan University, Safed 1311502, Israel

**Keywords:** extracellular vesicles (EVs), phytosomes, exosome, nanoparticles, microvesicle, cancer–cell crosstalk, plant-derived exosomes

## Abstract

Extracellular vesicle (EV) research has expanded substantially over the years. EVs have been identified in all living organisms and are produced and released as a means of intercellular communication or as a defense mechanism. Recently, nano-scaled vesicles were successfully isolated from edible plant sources. Plant-derived EVs, referred to here as phytosomes, are of a size reported to range between 30 nm and 120 nm in diameter, similar to small mammalian extracellular vesicles, and carry various bioactive molecules such as mRNA, proteins, miRNA and lipids. Due to the availability of many plants, phytosomes can be easily isolated on a large scale. The methods developed for EV isolation from mammalian cells have been successfully applied for isolation and purification of phytosomes. The therapeutic effects of phytosomes on different disease models, such as inflammation and autoimmune disease, have been reported, and a handful of studies have suggested their therapeutic effects on cancer diseases. Overall, the research on phytosomes is still in its infancy and requires more exploration. This review will narrate the anti-cancer activity and characteristics of phytosomes derived from edible plants as well as describe studies which have utilized phytosomes as drug delivery vehicles for cancer with the ultimate objective of significantly reducing the adverse effects associated with conventional therapeutic approaches.

## 1. Introduction

Extracellular vesicles (EVs) are subcellular structures enclosed by a phospholipid bilayer. EVs are ubiquitously produced by cells of all living organisms in homeostasis and disease, and mediate intercellular, inter-species and inter-kingdom communications [1,2]. EVs were initially nomenclatured by the International Society for Extracellular Vesicles (ISEV) based on their diameter size, with small EVs including 30–160 nm vesicles, microvesicles including 120–1000 nm vesicles, and large vesicles or apoptotic bodies including 1–6 μm vesicles [3,4,5]. EVs are typically classified in the literature as exosomes (<200 nm) or ectosomes (>200 nm). Exosomes are much more abundant in biological fluids than ectosomes, which include nano- and microscale EVs, e.g., microvesicles, microparticles, large vesicles and apoptotic bodies [6,7,8]. Small-sized ectosomes also exist and are similar to exosomes in their physical properties (e.g., density, charge, solubility, surface proteins, membrane lipid compositions, and shape), which creates challenges for separating pure exosomes from cell secretions [6,9,10]. However, the biogenesis of exosomes and ectosomes differ [7,11]; ectosomes are generated simply through a process of outward protrusion of the plasma membrane and budding out, and, as such, should have different cargo constituents and functions that are not the focus of this review [8,12]. Exosomes, on the other hand, have distinct pleiotropic functions that mediate near and long-distance intercellular regulatory processes in health and disease [13,14,15]. Most information on the function and biogenesis of exosomes is based on the study of mammalian EVs.

Exosomes are produced through the endocytic pathway in a process which starts with an inward invagination of the plasma membrane and formation of an intracellular endosome that contains cell-surface proteins and soluble components from the extracellular milieu [16,17]. The content of this early endosome is further enriched after it merges with other preformed endosomes, or by exchange of constituents from other organelles, such as the mitochondria, trans-Golgi network and endoplasmic reticulum. At this point, numerous inward invaginations of the early endosome result in the formation of multivesicular bodies (MVBs) that contain several vesicles deemed to be future exosomes [16,17]. The MVBs attach to the cytoskeleton network, and are then transported and dock at the luminal side of the plasma membrane. There, they secrete their contents into the extracellular space [7,8]. Due to uneven plasma membrane invagination during their biogenesis, the size, cargo and biological function of exosomes are heterogeneous, each bearing distinct abilities to induce complex biological responses in recipient cells [7,18,19]. The enclosed cargo content is also affected by the types and functional states of the releasing cells. The cargo is stably transferred from donor to recipient cells and, as such, carries potential for disease diagnosis and therapy [7,19].

Exosomes transfer endogenous effector or signaling molecules to adjacent or distant recipient cells under both physiological and pathological conditions. The cargo includes a broad spectrum of proteins (heat shock proteins (HSP), such HSP60, HSP70, and HSP90; biogenesis proteins, such as Alix and TSG 101; and membrane transport and fusion proteins, such as GTPases, annexins, and Rab proteins), lipids (cholesterol and ceramide), RNAs (miRNA, long non-coding RNA, tRNA, and mRNA) and DNAs (dsDNA, ssDNA, and mtDNA) [20]. Due to their ability to naturally encapsulate and transfer a broad range of molecules and proteins, exosomes have been utilized as a platform for targeted, high-efficiency delivery of therapeutic molecules [21,22,23], or other chemicals and biological drugs such as siRNA and microRNA to cancer cells [23,24,25,26,27]. Exosomes have also been analyzed for biomarkers for early diagnosis of diseases such as infectious diseases, autoimmune disorders, diabetes and several other types of cancers, [28,29]. Moreover, exosomes have been utilized as decoys against viral infections, such a SARS-CoV-2 [30], and their natural cargo, obtained from the donor cell, has been applied as a base treatment to inhibit angiogenesis and tumor growth [31]. Thus, the field of exosomes has opened new avenues and perspectives on how to approach complex diseases such as cancer.

Throughout the centuries, plants have been studied for their therapeutic compounds, and numerous extracts and phytochemicals have been purified and tested on various disease models. Many plant-derived components or molecules are used as drugs and therapies for conditions such as inflammation, autoimmune diseases, diabetes, cardiovascular disease and infections [32]. Recently, plant-derived exosome-like nanoparticles (PELNs) isolated from edible plants were found to be an abundant source of nanovesicles with remarkable therapeutic properties and minimal side effects [33,34,35]. These phytosomes display structure, density and roles in interspecies cellular communication similar to those of mammalian cell-derived exosomes [36,37]. While phytosome biogenesis is still being studied [38,39], the cargo has been analyzed and found to include bioactive and therapeutic phytochemicals [40,41,42,43]. Phytosomes have also been implicated in defense mechanisms against invading pathogens, particularly bacteria and fungi [42,43,44]. Due to their inherent biocompatibility and modulatory effects on aberrant cells, particularly cancerous cells, phytosomes present a range of advantages as carriers for therapeutic agents. Additional advantages include cost-effective isolation, efficient encapsulation of drugs and genetic material, stability in extracellular environments and the ability to be modified with specific target markers for precise cell targeting in vitro and in vivo. Consequently, phytosomes hold considerable promise as an effective delivery system for therapeutic agents [41,45], such anticancer drugs, siRNAs, miRNAs, PNAs and even poorly soluble natural compounds such as curcumin [39] and paclitaxel [46]. Despite the significant progress made in understanding cancer biology, the current treatments integrating the most innovative strategies from different disciplines still have certain limitations and challenges [47]. Nanomedicine is perceived as a discipline which can address such issues of conventional anticancer therapies [48].

A paramount study that paved the way for future studies on phytosomes involved their isolation from sunflower seed apoplastic fluid. The purified 50–200 nm phytosomes carried a protein similar to the human Rab11 GTPases, which was found to be involved in phytosome function, biogenesis and release [49]. Subsequent studies found that these vesicles are spherical structures enclosed by a lipid bilayer comprised of phospholipids and glycerol, similar to membranes in eukaryotes [34]. Their size was found to be similar to that of mammalian cell-derived EVs (100–1000 nm), as was their biological content, including RNAs and micro-RNAs [50]. Phytosome cargo contained agents with anti-inflammatory, immunomodulatory and regenerative effects and proved to be reliable products for drug and gene delivery as well [39,46,50]. In a report of phytosome isolation from tomato plants, approximately 1 kg of tomato fruits yielded 2.7 × 10^16^ (≤155 nm) and 3.8 × 10^16^ (≤110 nm) phytosomes [33], which was a significantly higher yield than that achieved with mammalian cell cultures [51]. The production scale of phytosomes from edible plants is abundant and sustainable, and the phytosomes can be easily purified and utilized for disease treatment and delivery of bioactive compounds to cells.

This review will focus on natural phytosomes that were successfully isolated and purified from fruits and vegetables of various edible plants, and their potential in treating cancer. We will narratively present studies, highlighting the anticancer activity of phytosomes and their utility for anticancer drug delivery.

## 2. Unique Characteristics of Phytosomes with Anticancer Activity

Safety, specificity and efficacy pose major obstacles to the development of new therapies. Unlike conventional treatments that fundamentally rely on pure compounds extracted from natural sources or on artificially manufactured products, which are often associated with side effects and can yield resistance if used in the long-term, phytosome-based drug delivery circumvents these issues [50]. Collectively, studies have demonstrated that phytosomes have no toxic effect on healthy cells, but rather, act on abnormal, transformed cells, e.g., cancer cells [52,53]. As a result, phytosomes are an emerging platform for plant-based therapy of cancer and chronic inflammation associated with cancer (Table 1) [52,53].

Phytosomes isolated from fingerroot (*Boesenbergia rotunda (*L.*) Mansf.*) demonstrated an apoptotic effect on colorectal cancer cells in vitro while showing no adverse effects on normal colon epithelial cells [54]. At a concentration of 50 µg/mL, these phytosomes induced toxicity after 24 h of incubation with cancer cells, but had no visible effect on normal colon epithelial cells. Phytosome uptake was shown to upregulate the pro-apoptotic genes encoding caspase 3, caspase 9, Bax and Bcl-2 and was strongly correlated with the release of reactive oxygen species (ROS) and cell death [54]. Phytosome internalization is suggested to partly involve the caveolae-mediated endocytosis and phagocytosis pathways. This was supported by pre-exposing cells to cellular uptake inhibitors, namely, the phagocytosis and pinocytosis uptake pathways. The study established a correlation between the anticancer effect and naringenin chalcone, pinostrobin and pinocembrin, three phenolic compounds highly abundant in the plant. These compounds were identified as the primary contributors to the observed anticancer activity. However, further molecular examinations are still necessary to substantiate the findings.

Furthermore, uptake of garlic (*Allium sativum*)-derived phytosomes by a human liver cancer cell line (HepG2) was found to depend on the CD98 receptor, as demonstrated by receptor-specific blockers [55]. This was further confirmed by trypsinizing phytosome surface proteins, which stopped the internalization of garlic phytosomes. The study also suggested a strong correlation between the presence of lectin family proteins on phytosome surfaces and CD98 receptor-mediated endocytosis. Interestingly, although they conferred an anti-inflammatory effect by downregulating proinflammatory factors, garlic phytosomes failed to show an anticancer or cell-modifying effect when applied to HepG2 cells. While these observations suggest an insignificant effect of the garlic phytosome cargo on liver cancer cells, the results may have been due to the low dose tested (i.e., micrograms). Nonetheless, the study provided insights into the specificity of garlic phytosome internalization by CD98 receptor-expressing target cells, which are typical of many types of cancers. Allicin, a well-defined bioactive anticancer agent found in garlic, has been extensively investigated to unveil its mechanism of action on cancer cells [56]. Therefore, conducting a rigorous evaluation of the intracellular cargo carried by garlic phytosomes and their ensuing interactions with HepG2 cells will be of paramount significance.

Phytosomes isolated from lemon juice (*Citrus limon* L.) were shown to impart antiproliferative and apoptotic effects on cells both in vitro and in vivo [53]. They were also demonstrated to reduce tumor size in a murine model inoculated with chronic myelogenous leukemia cells (LAMA84 cell line). The effective phytosome dose was reported to be 20 µg/mL. The antitumorigenic events were paralleled by the activation of TNF-related apoptosis-inducing ligand and TRIAL receptor, increased expression of Bad and Bax genes and decreased expression of antiapoptotic genes, namely, survivin and Bcl-xl. In vivo biodistribution analyses found that injected labeled *Citrus limon* phytosomes specifically accumulated in cancer cells; while some were absorbed by other organs, no significant anomalies were observed [53].

An important aspect that remained unexplored was the identity of the active compounds and biomolecules carried by phytosomes which could potentially be associated with the observed anticancer activity. Notably, *Citrus limon* is known to contain polymethoxylated flavones (PMFs), a class of bioactive compounds exclusive to citrus plants. Extensive research has demonstrated that these PMFs exhibit antimetastatic, anti-proliferative and antiangiogenic properties across various cancer models [57]. Therefore, comprehensive examination of the impact of these PMFs on cancer cells would have significantly strengthened the findings regarding the anticancer activity of *Citrus limon* phytosomes.

Sasaki et al. [58] successfully isolated phytosomes from the edible portion of corn (also called *Zea mays* or *Maize plant*) using a simple ultracentrifugation-based method that enabled large-scale extraction. The phytosomes demonstrated antiproliferative and apoptotic effects in vitro on a murine colon adenocarcinoma cell line (colon26) and were preferentially taken up by cancer cells compared with normal non-cancer cells, suggesting an affinity for the lipid rafts abundant on cancer cells [58,59]. They also demonstrated an anticancer effect on subcutaneous murine colon adenocarcinoma tumors in a syngeneic mouse model, with only a negligible impact on body weight. The involved apoptotic pathways or genes in were not studied. In addition, the cargo contained within corn phytosomes housing the potential anticancer biomolecules was not examined. However, a previous investigation reported on inhibited proliferation and induced apoptosis of colon adenocarcinoma and suggested the presence of the carotenoids zeaxanthin and lutein in corn [60]. Given this prior evidence, there exists a strong likelihood that corn phytosomes carry these bioactive compounds, which may be closely associated with the observed antineoplastic effects. The presence of zeaxanthin and lutein in corn phytosomes could potentially play a significant role in the elicitation of anticancer responses. Conducting an in-depth investigation to confirm the presence and quantity of these biomolecules within corn phytosomes would enhance our understanding of the underlying mechanisms driving the observed antineoplastic effects. Notably, concentrations of 500 µg/mL and 1000 µg/mL of phytosomes exhibited potent anticancer effects on a colon cancer cell line in vitro and in vivo, respectively [58,59]. In a subsequent study, Sasaki assessed the biodistribution of corn phytosomes and phytosomes with a polyethylene glycol (PEG)-modified surface [59]. Intravenously injected phytosomes predominantly accumulated in mouse spleens and livers, whereas PEG-phytosomes exhibited lower accumulation in these organs. Importantly, the surface modification prolonged phytosome circulation in the bloodstream and their preferential accumulation in the tumor site.

Cannabis (*Cannabis sativa* L.) is an edible plant which can be used to prepare cakes, sweets, chocolates and drinks. While it is less commonly used for medicinal purposes, its phytosomes have demonstrated promising anti-cancer potential. Researchers utilized differential centrifugation and a sucrose gradient to isolate and purify cannabis phytosomes that contained low levels of tetrahydrocannabinol (THC), the primary psychoactive compound in this plant family, and a high CBD content (referred to as H.C-EVs). The isolated phytosomes demonstrated dose- and time-dependent anti-cancer effects on HepG2 and Huh-7 liver cancer cell lines. A dose as low as 25 µg/mL phytosomes induced apoptosis of cancer cells, while 100 µg/mL caused a 50% reduction in cancer cells viability. No cytotoxic effects were observed on non-cancerous cells such as the HUVEC line [61]. The apoptotic effects in vitro were associated with the upregulation of proapoptotic markers bax, CASP3, and CASP9 and downregulation of the antiapoptotic gene bcl-xl. However, the assessment of cannabis phytosome uptake by cells would have been pivotal to provide more definitive evidence on their activity. Residual CBD in the sample even after phytosome purification could have been present and may have triggered effects on the cancer cells as CBD is known to interact with the CB2 receptor and induce apoptosis in tumor cells [61,62]. Overall, the isolation of phytosomes from cannabis plants and their CBD contents is novel and intriguing as much of the cannabinoids research to date focuses on the activity of phytochemicals and their effects on biological systems.

Tea leaves (*Yongchuan Xiuya*), referred to as TLNTs, have been found to possess potent anticancer properties that can be attributed to phytosomes carrying well-documented anti-cancer polyphenols and flavonoids, such as epigallocatechin-3-gallate (EGCG), vitexin-2-*O*-rhamnoside, vitexin, myrice-tin-3-*O*-rhamnoside, kaempferol-3-*O*-galactoside and myricetin [63]. Tea leaf phytosomes intravenously or orally administered to breast cancer-bearing mice accumulated at the malignant site and inhibited tumor growth after only 5 h. In vitro studies showed that the phytosomes were taken up into the cytoplasm of cancerous cells and induced the release of ROS, which resulted in cell damage. Intravenous injection of high doses of phytosomes was found to elevate liver markers, while there was no evidence of toxicity or side effects following oral administration, suggesting that oral treatment with phytosomes is a safer route. Interestingly, phytosomes augmented the number of favorable bacteria inside the gastrointestinal tract (GIT), while reducing pathogenic bacteria.

Phytosomes from apple fruits (*Malus domestica* sp. *Variety Golden Delicious*), referred to as ADNVs, have been shown to confer anti-inflammatory effects on proinflammatory macrophages, particularly of the polarized M1 phenotype [64]. Phytosomes isolated from macerated apple pulp had a size range of 90–180 nm. Reduced levels of IL-1b and IL-8, as well as upregulation of mir-146 and miR-125a, were documented following exposure of THP-1 cells to ADNVs. Fluorescence microscopy captured the uptake of phytosomes into the cytoplasm of THP-1 cells. Characterization of the endocytic pathway, along with analysis of the cargo contents, could have provided clarity regarding the mechanism of phytosome action. In this regard, further research will be required to define the distinct properties of apple phytosomes. A major insight from the study was the well-documented phenomenon of macrophage exosome-mediated communication with cancer cells [65], which emphasizes the potential of blocking the tumorigenic influence of neighboring cells.

In a study focused on the cross-talk between phytosomes from edible plants and mammalian gut cells, such as intestinal macrophages and gut stem cells [41], it was found that certain phytosomes, such as those derived from ginger root, can modulate the release of anti-inflammatory genes, specifically heme oxygenase 1 (HO1), IL-10 and IL-6, and stimulate antioxidant genes in cultured macrophages. Some orally administered phytosomes were internalized by stem cells and macrophages in the gut and intestine of a murine model. The study outcomes suggest that phytosomes from distinct sources can withstand low pH, which may render them particularly useful as a platform for drug delivery in acidic environments, such as in the case of gastrointestinal carcinomas.

**Table 1 cells-12-01999-t001:** Anticancer effects of edible plant-derived phytosomes.

Phytosome Source	Isolation Method	In Vitro Effect	In Vivo Effect	Mechanism	References
Fingerroot (*Boesenbergia rotunda (*L.*) Mansf.*)	Fingerroot blended without addition of other liquids. Juice was filtered, ultracentrifuged and passed through a size-exclusion chromatography column.	Phytosomes induced selective cytotoxic and apoptotic effects on cancer cells but not on normal colon epithelial cells.	-	Disruption of intracellular redox homeostasis and induction of cell apoptosis.	[54]
Garlic (*Allium sativum*)	Garlic was homogenized with a blender, in cold PBS. Juice was subjected to a multistep centrifugation procedure, ending in ultracentrifugation. The phytosome pellet was washed and resuspended in PBS.	Phytosomes were internalized and induced an anti-inflammatory effect but not an anti-cancer effect on HepG2 cancer cells.	-	The phytosomes were internalized via the CD98 receptor located on HepG2 cells. The receptor is a glycoprotein rich in mannose motifs that bind phytosome surface proteins and lectins.	[55]
Lemon juice (*Citrus limon* L.)	Fruits were manually squeezed and juice was sequentially centrifuged. The supernatant was filtered and then ultracentrifuged and the phytosomes from the pellet were purified on a 30% sucrose gradient.	Phytosomes inhibited cancer cell growth without affecting normal cells.	Phytosomes suppressed growth of subcutaneous tumors in NOD/SCID mice through TRAIL-mediated apoptosis and inhibition of angiogenic processes.	Phytosomes induced cell death of cancer cells by triggering TRAIL-mediated apoptosis. TRAIL selectively induced apoptosis of cancer cells without affecting normal cells.	[53]
Corn (*Zea mays* or *Maize plant*)	A homogenate of a blended edible portion of corn with water was prepared, then step-centrifuged to remove debris. The supernatant was filtered and ultracentrifuged. The pellet then resuspended in PBS to obtain phytosomes.	Phytosomes selectively inhibited proliferation of colon26 cancer cells.	Phytosomes significantly suppressed the growth of subcutaneous colon26 tumors in mice, with no side effects such as body weight loss.	-	[58,59]
Cannabis (*Cannabis sativa* L.)	A homogenate of a complete flower head of cannabis plant with cold PBS was sequentially centrifuged to remove debris, and the EV-enriched supernatant was filtered and ultra-centrifuged. The pellets were resuspended in PBS and resolved on a sucrose density gradient to further purify isolated phytosomes.	Phytosomes strongly decreased viability of two hepatocellular carcinoma cell lines (HepG2 and Huh-7), in a dose- and time-dependent manner. The phytosomes had no significant effect on the normal growth of HUVECs.	-	The phytosomes induced cell death by activating the mitochondria-dependent CASP and CASP9 pathways and upregulating proapoptotic markers bax while downregulating antiapoptotic gene bcl-xl. Phytosome uptake studies were not performed.	[61]
Tea leaves (*Yongchuan Xiuya*)	Fresh tea leaves were homogenized with PBS in a blender. Juice was sequentially centrifuged to remove debris. The supernatant was then resolved via sucrose density gradient ultra-centrifugation and phytosomes were collected from the 30/45% sucrose interface.	Phytosomes were internalized and decreased the viability and enhanced the cytotoxicity of three breast cancer cell lines.	Intravenously and orally administered phytosomes decreased mammary gland tumor size in nude mice.	Phytosomes increased intracellular reactive oxygen species (ROS) levels, which damaged mitochondria, and triggered cell cycle arrest and apoptosis.	[63]
Apple fruits (*Malus domestica* sp. *Variety Golden Delicious*)	The pulp of apple fruits was homogenized, and subjected to a series of centrifugations to remove debris. Juice supernatant was filtered and further centrifuged to remove smaller debris. The supernatant was ultracentrifuged and the phytosome-containing pellet was resuspended in PBS.	Treatment of type I macrophages with phytosomes resulted in decreased expression of IL-1b and IL-8.	-	Phytosomes directly communicated with the immune system, and switched it to an anti-inflammatory mode.	[64]

## 3. Phytosomes as Drug Delivery Vehicles for Cancer Treatment

Research pertaining to the cargo contents of phytosomes with anticancer properties has established a novel framework for nanomedicine and therapeutic nano-delivery. Several studies have successfully demonstrated use of phytosomes isolated from various sources to encapsulate or incorporate drugs and biomolecules. A study comparing methotrexate (MTX) as a monotherapy and methotrexate conjugated onto grapefruit phytosomes (referred to GMTX) found significantly different adverse effects and therapeutic benefits for the two [66]. Oral administration of GMTX reduced inflammation in an induced colitis murine model. In parallel, a decline in inflammatory markers produced by colonic macrophages, such as TNF-α and IL-1β, was measured at the mRNA and protein levels, as well as a reduction in the expression of the neutrophil chemokine KC (CXCL1). In contrast, MTX monotherapy induced notable tissue damage and a reduction in E-cadherin expression. The study suggested a synergetic effect between phytosomes and the incorporated therapeutic agents such as MTX.

A group of investigators assessed the feasibility of utilizing inflammatory cell-derived membranes to coat grapefruit phytosomes to generate pseudo-inflammatory grapefruit phytosomes (referred to as IGNV), with the aim of mitigating off-target delivery [67]. When compared with uncoated phytosomes, the modified phytosomes, which were bound to membranes bearing chemokine receptors derived from activated EL4 T-lymphocytes cells, exhibited greater attraction to inflammatory sites, particularly to regions with high chemokine levels. Additionally, in mice challenged with lipopolysaccharide (LPS) or bearing CT26 colon cancer or 4T1 breast cancer, IGNV exhibited greater attraction to regions with aberrant release of inflammatory signal molecules. Notably, IGNV accumulation was more prominent at tumor sites compared with LPS-induced inflammation sites. The cxcr2 and LFA-1 ligand were identified as the principal ligands responsible for this outcome as their elimination led to a significant reduction in vesicle migration. Furthermore, the researchers demonstrated the superior effectiveness of therapeutics loaded into IGNVs; notably doxorubicin, a known chemotherapeutic, and curcumin displayed a more pronounced effect on abnormal cells when bound to phytosomes and localized at the tumor site as compared with other locations [67].

A strategy under development to disrupt the activity of cancerous cells involves the targeted delivery of micro-RNA (miRNA) molecules, which are noncoding sequences that regulate the functionality of messenger RNA [68]. Phytosomes were shown to effectively deliver miRNA molecules to neoplastic sites. For example, phytosomes from grapefruits were demonstrated to deliver mir-18a to liver malignancies and inhibit metastasis [67]. Phytosomes encapsulating mir-18a prompted an elevation in M1 macrophage activity, resulting in the secretion of IFNγ and IL-12, alongside a decline in TGFβ- and IL-10-releasing M2 cells, leading to stimulation and recruitment of natural killer cells and T-cells to the malignant site. The phytosomes also induced the expression of IFNγ, a pivotal signaling molecule implicated in cancer progression. The cascade of events initiated within the cancer site by mir-18a-loaded phytosomes resulted in a reduction in tumor size and suppression of metastasis.

GNVs generated via the assembly of grapefruit lipid extracts were designed to incorporate lipids to improve targeting, phytosome size uniformity, and chemotherapeutic drug loading [69]. Following intranasal administration, GNVs bearing mir17 reached the neoplastic region of a murine brain tumor model and were internalized by the cancerous cells, which impeded their proliferation through the downregulation of MHCI ligand on glioma and was correlated with the induction of natural killer cells (NK). A biodistribution analysis found that the nanovectors accumulated in other regions of the murine brain without eliciting adverse effects, which represents an advantage over synthetic liposome-based therapies that carry certain risks. Integration of polyethyleneimine (PEI), a transfection agent, increased mir17 levels in the vesicles and subsequent mir17 accumulation within the brain tumor cells. Additionally, conjugation of folic acid to nanovectors improved targeting to neoplastic brain cells in vivo, due to the upregulation of the folic acid receptor on many cancer cells [70].

Phytosomes isolated from cabbage, colloquially termed Cabex and Rabex, successfully conveyed therapeutic agents without undergoing any morphological changes. This suggested the significant potential of cabbage-derived phytosomes as competent carriers for substantial drug molecules [71]. In vitro assays demonstrated that when loaded with doxorubicin, the phytosomes, at a concentration of 1 × 10^9^ phytosomes/mL, exerted oncolytic effects on SW480 neoplastic cells and reduced cell viability to a range of 50–60%. The study also demonstrated that phytosomes can encapsulate ample quantities of free miRNA and deliver them to target cells without a major loss in miRNA in the process. This observation was supported by successful incorporation of fluorescent dye-conjugated antisense DNA oligonucleotides and miRNA into the phytosomes and tracking their uptake into colon cancer cells. Additionally, phytosome uptake by healthy HaCaT and HDF cell lines had no effect on homeostatic morphological characteristics and even promoted propagation. These results represent an invaluable foundation for high-scale, cost-efficient isolation of phytosomes from cabbage. Regardless, further exploration is warranted to decipher the specificity, biodistribution, cellular uptake pathways and the potential role of cabbage-derived phytosomes in alleviating the deleterious effects associated with doxorubicin administration.

Lemon-derived phytosomes modified with carboxyl groups of heparins (HRE) and loaded with cisplatin, taxol or doxorubicin [72] demonstrated significant antiproliferative and apoptotic effects on multidrug resistant cervical cancerous cells (SKOV-3 cells), compared with the free drugs, by reducing the production of ATP and increasing ROS release. In a murine model, the anticancer drugs accumulated in tumor cells, while the free drugs were hardy detectable, indicating that the limon phytosomes overcame the cell mechanism of resistance. Biodistribution assessment showed homing of the majority of phytosomes to the neoplastic site. Inflammation analysis found no disturbance of the systematic homeostasis in the treated mice. The uptake of the modified limon phytosomes was associated with caveolin-mediated endocytosis and proved to be less dependent on clathrin-mediated endocytosis and macropinocytosis.

As phytosomes are biocompatible, abundant and cost-effective, and can be administered intravenously or orally, it is clear by now that they can serve as a favorable means of delivering anti-cancer drugs. Such an approach may circumvent the side effects associated with the clinical applications of well-known chemotherapeutic anti-cancer drugs. Taxanes, such as the naturally derived paclitaxel (PTX, brand name Taxol^®^) and the semi-synthetic analogue docetaxel (DTX, brand name Taxotere^®^), are FDA-approved and the most effective and commonly used anti-cancer drugs [73]. Despite their anti-proliferative and apoptotic effects on cancer cells, taxanes impart dose-dependent toxicities, exhibit poor solubility and low selectivity for target tissue and induce hypersensitivity reactions [74]. Taxane formulations are generally prepared with certain solvents due to their low aqueous solubility and are consequently not well tolerated, causing severe side effects. In recent studies, PTX and DTX conjugates have been prepared to reduce toxicity, improve efficiency and broaden the therapeutic window [75,76]. These side effects may be easily overcome by their encapsulation within phytosomes and intracellular delivery by endocytosis, subsequently enhancing their inhibitory effect on cancer cells and lowering their adverse effects on normal cells. Indeed, orally administered mammalian-derived exosomes encapsulating PTX exhibited strong anti-lung cancer activity and a favorable safety profile [46]. Similarly, in our recent study, we utilized lipofectamine, a lipid-based, liposome-forming transfectant, to complex with peptide nucleic acid (PNA)-based antisense. Through the endocytosis pathway, the liposomes enhanced the intracellular delivery of the PNAs and, thereby, inhibited the proliferation and induced apoptosis of pancreatic cancer cells [77]. Since lipofectamine is only used for research purposes for transfection of nucleic acids into eukaryotic cells, future studies will focus on the delivery of PNA-based treatments by means of oral or intravenous phytosome formulations.

## 4. Discussion

Nanomedicine, with a specific focus on phytosomes, is proving a revolutionary avenue for innovative therapeutic anticancer strategies. Studies in this field are swiftly gathering momentum due to the advantageous features of phytosomes as a platform to deliver bioactive cargo to specific cell targets and, thereby, alter their functional behavior. The cumulative evidence relating to phytosome subgroups isolated from edible plants and from meticulous investigation of their cargo has blazed the trail for exploiting them for cancer treatments by leveraging their natural cargos, as well as establishing their utility for drug delivery. At present, many efforts are focused on the development of mammalian-derived EVs and artificial nanoparticles as novel therapies for various diseases, with emphasis on tumoricidal activities. The development of mammalian-derived EV therapy fundamentally relies on the isolation of crude EVs from cell environments, such as cell culture supernatants, biological fluids and intercellular spaces within tissues, followed by purification of subgroups of vesicles using distinct isolation techniques and screening for EV-associated protein markers such as tetraspanins CD63, CD81 and CD9 [78]. On the other hand, artificial nanoparticles are chemically synthesized and designed to improve the pharmacokinetics or pharmacodynamics of drugs, increase their bioavailability and reduce associated side effects [79,80]. Despite being at an early developmental stage, the potential of phytosomes in the realm of cancer nanotherapy is being increasingly recognized. Phytosomes are particularly attractive due to their scalability, pronounced specificity towards tumor cells compared with healthy cells, unique cargo compositions and engagement of the endocytosis pathways of mammalian cells. The most distinctive feature of phytosomes is their minimal toxicity to cells and organs, likely due to the fact that they are an integral part of our daily diet, as substantiated by their frequent presence in various dietary plants [81]. As such, adoption of phytosomes over other alternatives in nanomedicine holds several advantages as their isolation is cost-effective and requires low-cost initial raw biomaterials and facilities. Their extraction generally involves homogenization of plants and subsequent simple processing procedures [35]. In contrast, EVs derived from mammalian cells require extensive cell line or primary cell culture expansions, or viable tissues, necessitating substantial amounts of conditioned medium, reagents, enzymatic digestions and other instruments to yield a high volume of purified EVs [51,82]. In addition, EVs isolated from human tissues are generally limited by tissue availability and regulatory guidelines, rendering this approach commercially impractical [80,83]. The constant need for adjustments in cell culturing—the main source of EV production—due to regulatory requirements, also constitutes the primary challenge of mammalian-derived EV therapies. Fetal bovine serum (FBS) is a reagent vital for maintenance of stable cell cultures. However, regulatory authorities are progressively compelling the biotechnology industry to explore serum-free alternatives, primarily driven by concerns surrounding immunogenicity, animal welfare, and the presence of prion proteins and other contaminants. These adjustments could potentially restrict the use of certain cell types as not all serum-free media formulations are suitable for all cell origins. Consequently, this limitation may hinder large-scale EV production and increase costs [84,85].

Phytosomes were found to be highly stable under various environmental conditions, such as different temperatures and pH values. The development of treatments or therapeutic agents normally requires their evaluation under the conditions of their planned route of administration. Phytosomes derived from distinct plant sources were found to tolerate and maintain their biophysical characteristics and potent activities in low-pH environments [41,86], exhibit resistance to digestive enzymes [41,45], and to elicit no adverse effects in vitro or in vivo. There are a number of ongoing phase I clinical trials utilizing phytosomes for treatment of diseases that will help elucidate their safety, dosing range, side effects and drug delivery capacities (https://clinicaltrials.gov, accessed on 21 May 2023, NCT 01294072, NCT 01668849, NCT 03493984). Phytosomes can also cross biological barriers, including the blood-brain barrier, intestine, skin and many others, but not the placental barrier (from mother to fetus) [34,70,87]. These biophysical characteristics will be beneficial in the development of therapeutic agents for many cancer diseases, as indicated in Figure 1.

Despite the promising potential of phytosomes in the realm of nanomedicine, there are noteworthy challenges. Phytosomes remain to be comprehensively characterized, with particular emphasis on the identification of surface and internal biomarkers. Only a handful of studies have been successful in identifying a limited set of markers in plants that are typically found in mammalian-derived EVs. For example, Rab11 GTPases were identified in sunflower seed phytosomes [41,45], while actin, heat shock protein 70, GADPH and S-adenosyl-homocysteine hydrolase were detected in phytosomes derived from olive pollen grains [88]. Furthermore, regardless of whether the delivered drugs are derived from natural sources or synthesized, when compared with conventional medical interventions, the utility of phytosomes needs to be further established to optimize reproducibility, dose toxicity, specificity, structural integrity and cost-effectiveness [80]. It will be imperative to identify isolation techniques that ensure reproducible and pure yield. Research related to phytosomes has predominantly utilized methodologies previously employed for the separation of EVs from mammalian cells, with ultracentrifugation commonly regarded as the standard benchmark technique. Agricultural practices must also be considered as they can introduce influences, including genetic modification, pesticide exposure and climate change. These, in turn, can affect the intracellular contents of plants and production of phytosomes and their cargos [89]. Hence, it is necessary to have a controlled source of raw plant material to maintain consistency and reproducibility.

In summary, the emerging subdomain of phytosomes within the broader field of EVs holds as much potential as the initial discovery of exosomes isolated from reticulocytes four decades ago [90,91]. Therefore, we postulate that investments in this field are likely to be fruitful and can significantly advance the development of antineoplastic nanomedicine.

## 5. Conclusions

Nanomedicine researchers have recently tapped into the potential of agents at the nanometer scale to develop products for disease treatments and delivery of drugs to specific targets in the human body. Nanomedicine technology can make significant contributions to the health industry, medicine, and pharmaceutics. In particular, phytosomes are associated with advantages such as cost-effective production, availability and simple isolation. Their ability to cross biological barriers, be taken up by cells and transfer cellular information, are also of crucial relevance. Phytosomes can penetrate mammalian cells in vitro, confer selective cytotoxic effects by inhibiting growth and inducing apoptosis in cancer cells only and strongly suppress tumor growth in animal models. The minimal cytotoxicity of phytosomes to homeostatic normal cells emphasizes their usefulness in drug delivery. Drugs encapsulated within phytosomes, such as doxorubicin, curcumin and micro-RNA molecules, have been shown to affect cancer cells, reduce tumor size, suppress metastasis, and have improved targeting. Well-known chemotherapeutic agents that can hamper tumor growth, such as paclitaxel and docetaxel, can be delivered via phytosomes to increase their bioavailability and decrease their side effects.

Research on phytosomes is still in its early stages; future studies will be essential to improve isolation procedures, and downstream analysis will be necessary to obtain reproducible and effective phytosomes. Biomarkers specific to phytosomes remain to be identified and will greatly help in their identification and reproducible isolation. Further investigations to establish the anticancer effect of phytosomes and to better characterize their cargo content as well as the identity of their membrane proteins, will shed light on their selectivity to cancer cells and help in establishing drug-loading protocols. Such future studies may provide insights as to why herbivorous mammals, including humans on plant-based diets, are more protected against cancer.

## Figures and Tables

**Figure 1 cells-12-01999-f001:**
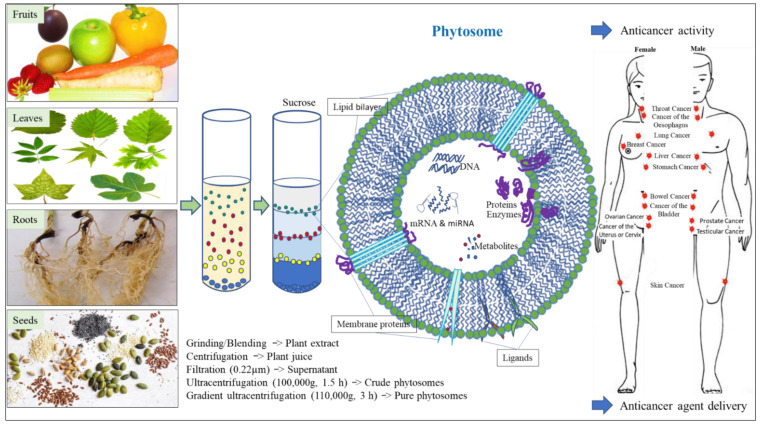
Application of edible plant-derived phytosomes for cancer treatment. Representative figure of phytosome showing its general structure and composition.

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
