# Peer review of "Nanoscale Phytosomes as an Emerging Modality for Cancer Therapy"

_cells, 2023, doi:10.3390/cells12151999_

Round 1
Reviewer 1 Report
The authors presented a review on the use of phytosomes (plant exosomes) in cancer diseases as drug delivery systems and anti-cancer activity. The manuscript contains information on the classification and identification of EVs, the isolation of phytosomes from plants and their potential in cancer. However, the following points need an improvement before a reevaluation for publication.
1. Line 82-85, the authors describe their subjects. After that paragraph, before the other sentence begins, a title should be added.
2. In the manuscript, phytosomes described as plant-derived EV’s, plant exosomes, plant-derived exosome-like nanoparticles. Although they all mean the same thing, there are differences between their sizes. It is more appropriate to use a single definition, so that it is more understandable for the readers.
3. The image quality of Figure 1 should be increased and the texts in the image should be in a more legible size.
4. If the entire Figure 1 does not belong to the authors, the source should be cited.
5. In section 3 authors use a limited reference, considering a use of more references relevant with the topic will enhance the strength of the section.
6. A bibliometric analysis will add a strength to the discussion considering the background of the existing literature and the future perspectives about phytosomes.
7. Adding a swot analysis like vision will help the authors to add a progress in their discussion. Clear and detailed discussion of the pros and cons is critical to add the authors’ vision to the topic.
Reviewer 2 Report
The Authors present very comprehensive review on the current state-of the art of phytosome production and utilization. The MS should be accepted as is.
Reviewer 3 Report
This kind of review has been published several times before (https://doi.org/10.2174/1389450118666170508095250, https://doi.org/10.1016/j.heliyon.2023.e16561 ), especially the second one, which is the newest paper in 2023. It has a comprehensive summary of phytosomes in treating cancer, including type, fabrication, anti-cancer activity, and drug delivery of phytosomes. So, what’s the novelty of your manuscript?
Following are the other suggestions:
1. What are the active ingredients in phytosomes to help their anti-cancer effect?
2. What’s the apparent difference between phytosomes and mammalian small extracellular vesicles, such as high yield? Please make a list of it in a single paragraph.
3. Line 99-100, the authors mentioned the phytosome can selectively target cells in vitro and in vivo, can you please offer more details? How to achieve this?
4. In Table 1, I suggest adding a publication year row.
5. Please supplement figures (scheme illustration) below parts 2 and 3, and may also add a table in part 3 just like table 1 to make a list of its carrier's usage.
The usage of definite and indefinite articles has minor problems.
Round 2
Reviewer 3 Report
Following question 3, I noticed in section 3, the phytosome achieves targeting only based on the link with other targeted ligands or molecules such as inflammatory cell-derived membranes or folate acid, but not phytosome itself, am I correct?
However, lines 99-100 will mislead the reader that phytosome itself has a targeting ability through some mechanism.
The grammar problem I mentioned last time was not solved. Some gerunds and infinitives are also misused such as line 24 should be"suggested utilizing".
